

# Neural and endocranial anatomy of Triassic phytosaurian reptiles and convergence with fossil and modern crocodylians

Stephan Lautenschlager[1] and Richard J. Butler[2]

[1] School of Earth Sciences, University of Bristol, Bristol, United Kingdom
[2] School of Geography, Earth and Environmental Sciences, University of Birmingham, Birmingham, United Kingdom

## ABSTRACT

Phytosaurs are a clade of large, carnivorous pseudosuchian archosaurs from the Late Triassic with a near cosmopolitan distribution. Their superficial resemblance to longirostrine (long-snouted) crocodylians, such as gharials, has often been used in the past to infer ecological and behavioural convergence between the two groups. Although more than thirty species of phytosaur are currently recognised, little is known about the endocranial anatomy of this clade. Here, we describe the endocranial anatomy (including the brain, inner ear, neurovascular structures and sinus systems) of the two non-mystriosuchine phytosaurs *Parasuchus angustifrons* (=''*Paleorhinus angustifrons*'') and *Ebrachosuchus neukami* from the Late Triassic of Germany based on digital reconstructions. Results show that the endocasts of both taxa are very similar to each other in their rostrocaudally elongate morphology, with long olfactory tracts, weakly demarcated cerebral regions and dorsoventrally short endosseous labyrinths. In addition, several sinuses, including large antorbital sinuses and prominent dural venous sinuses, were reconstructed. Comparisons with the endocranial anatomy of derived phytosaurs indicate that Phytosauria is united by the presence of elongate olfactory tracts and longitudinally arranged brain architecture—characters which are also shared with Crocodyliformes. However, a substantial morphological variability is observed in the cephalic and pontine flexure and the presence of a pineal organ across the different phytosaur species. These results suggest that the endocranial anatomy in Phytosauria generally follows a plesiomorphic pattern, with moderate variation within the clade likely resulting from divergent sensory and behavioural adaptations.

## INTRODUCTION

Phytosaurs are a group of fossil archosauriform reptiles commonly found in Upper Triassic (c. 235–202 Ma) sediments in North America and Europe, and less commonly in other regions such as India, Africa, East Asia, Madagascar and South America (*Stocker & Butler, 2013*). Phytosaurs have usually been regarded as the earliest diverging group within

Corresponding author
Stephan Lautenschlager,
glzsl@bristol.ac.uk

the crocodylian stem-lineage Pseudosuchia (*Sereno*, *1991*; *Parrish*, *1993*; *Brusatte et al.*, *2010*; *Ezcurra*, *2016*), which together with Avemetatarsalia (pterosaurs, dinosaurs, birds) form the clade Archosauria. One recent phylogenetic dataset has recovered Phytosauria as a monophyletic clade just outside of and as a sister taxon to Archosauria (*Nesbitt*, *2011*), although this result has not been supported by a recent comprehensive revision of the phylogeny of early archosauriforms (*Ezcurra*, *2016*). Morphologically, phytosaurs resemble extant crocodylians, particularly longirostrine morphotypes such as gharials. Members of both groups possess large elongate skulls equipped with conical teeth, rows of sculptured osteoderms covering the axial and appendicular skeleton, and are characterised by a quadrupedal, sprawling gate (*Westphal*, *1976*). Evidence from taphonomy and ichnofossils suggests that, similar to crocodylians, phytosaurs were generally aquatic or semi-aquatic (*Buffetaut*, *1993*; *Renesto & Lombardo*, *1999*), but were also capable of terrestrial locomotion (*Parrish*, *1986*). Although phytosaurs and the earliest fossil crocodylians are significantly separated temporally (by about 100 million years) and phylogenetically, gross morphological similarities between the two groups have often been cited as evidence for ecological and behavioural convergence (*Camp*, *1930*; *Anderson*, *1936*; *Hunt*, *1989*; *Hungerbühler*, *2002*; *Witzmann et al.*, *2014*). However, phytosaurs are defined by a number of osteological characters that differentiate them from crocodylians, such as an elongate premaxilla, the caudal position of the external nares (which is placed close to the orbit in phytosaurs, rather than at the tip of the rostrum), and the absence of a secondary palate. Convergence in the form of a longirostrine skull shape has occurred numerous times throughout the evolution of pseudosuchian archosaurs (*Brochu*, *2001*); presumably as an adaptation to a specific habitat and diet (e.g., piscivory) (*Pierce, Angielczyk & Rayfield*, *2008*). However, the extent to which this osteological convergence is also reflected in soft-tissue structures, such as the endocranial anatomy, remains unclear. Neuroanatomical adaptations to a specific ecology or behaviour in phylogenetically divergent groups as drivers for morphological similarities have been suggested in avemetatarsalian ("bird-line") archosaurs (*Witmer et al.*, *2003*).

In the past, research on phytosaurs has largely focussed on comparative osteology, taxonomy and phylogenetic relationships. Due to their near-global geographic distribution but restricted temporal distribution phytosaurs have been used as index fossils in biostratigraphy. In comparison, the reconstruction and study of the endocranial anatomy of phytosaurs has received little attention (e. g., *Cope*, *1888*; *Case*, *1928*; *Mehl*, *1928*; *Camp*, *1930*; *Chatterjee*, *1978*). Most recently, *Holloway, Claeson & O'Keefe* (*2013*) described a digital endocast of the derived mystriosuchine phytosaur *Machaeroprosopus mccauleyi* (="*Pseudopalatus mccauleyi*") in order to evaluate the evolution of sensory systems in archosaurs.

Here, we describe the endocranial anatomy (including the brain, inner ear, neurovascular structures and sinus systems) of the two non-mystriosuchine phytosaurs *Parasuchus angustifrons* (="*Paleorhinus angustifrons*") and *Ebrachosuchus neukami* (*Butler et al.*, *2014*) (see *Kammerer et al.*, *2016*, for recent taxonomic revisions) (Fig. 1) based on digital reconstructions. Further comparisons are made with existing reconstructions for other phytosaurian and crocodylian taxa.

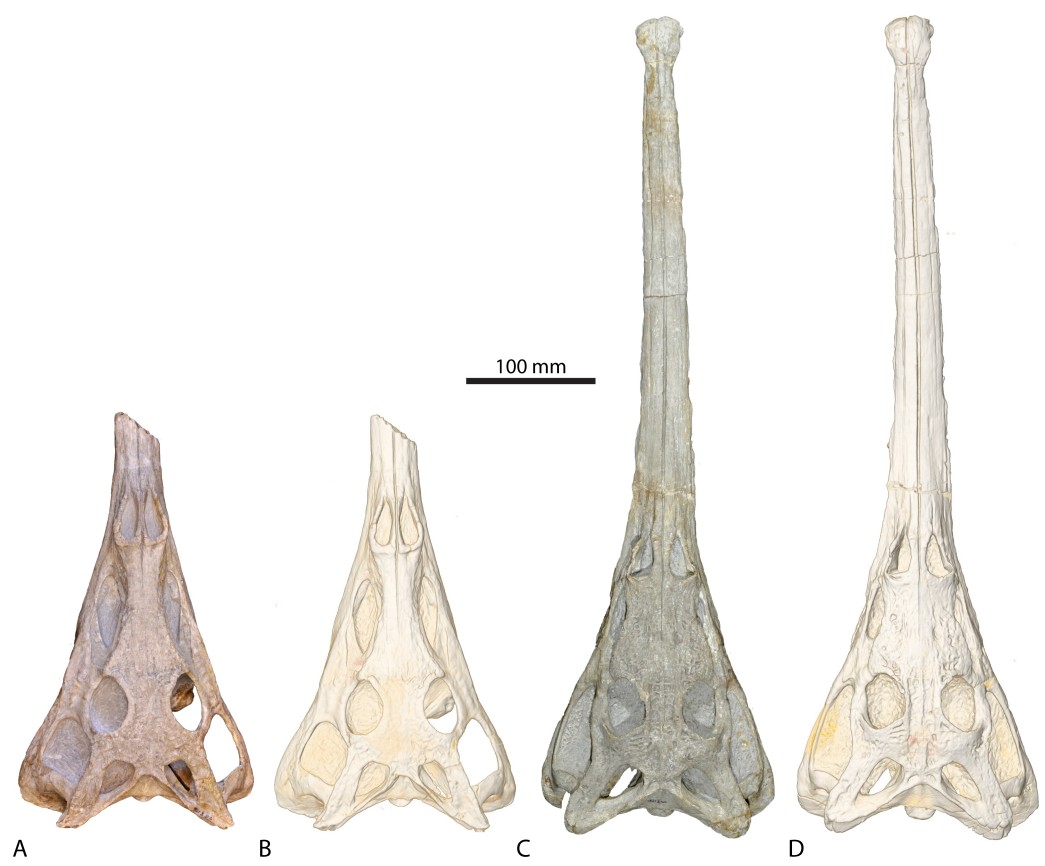

**Figure 1** **Studied phytosaurian taxa.** Physical specimen (A, C) and digital representation (B, D) of (A, B) *Parasuchus angustifrons* (BSPG 1931 X 502) and (C, D) *Ebrachosuchus neukami* (BSPG 1931 X 501).

## MATERIALS AND METHODS

### Specimens

The studied specimens consist of the holotypes of *Ebrachosuchus neukami* (BSPG 1931 X 501; Bayerische Staatssammlung für Paläontologie und Geologie, Munich, Germany) and *Parasuchus angustifrons* (BSPG 1931 X 502) (*Kuhn*, *1936*; *Butler et al.*, *2014*). The skull of *Ebrachosuchus neukami* is fully articulated and complete and preparation work has removed most of the sandstone matrix, with the exception of most internal cavities, which remain filled with matrix. The skull of *Parasuchus angustifrons* is articulated and mostly complete apart from the premaxilla, of which only a small portion immediately rostral to the external nares is preserved. Sandstone matrix remains within most of the internal cavities. Both of the skulls have undergone some plastic deformation, primarily in the form of dorsoventral compression, but are otherwise remarkably well preserved, with minimum transverse distortion. Brittle deformation and fracturing are largely absent.

### CT scanning and digital reconstruction

The holotypes of *Ebrachosuchus neukami* (BSPG 1931 X 501) and *Parasuchus angustifrons* (BSPG 1931 X 502) were scanned at the Klinikum rechts der Isar (Munich) using a Siemens SOMATOM Sensation 64 CT scanner. Datasets consisted of 1,634 slices ($512 \times 512 \times 1,634$

pixels, 0.6 mm voxel size) for *Ebrachosuchus neukami* and 809 slices (512 × 512 × 1,634 pixels, 0.6 mm voxel size) for *Parasuchus angustifrons*. CT data sets are deposited with the specimens in the BSPG collections and on Figshare (10.6084/m9.figshare.3443963; 10.6084/m9.figshare.3443960).

The respective CT data files were imported into Avizo 7.0 (Visualisation Science Group) for image segmentation and digital reconstruction. Anatomical structures of interest (endocasts, endosseous labyrinths, neurovascular and sinus structures) were labelled using Avizo's segmentation editor. The magic wand tool was used where possible to perform the segmentation semi-automatically. In regions with poor contrast between matrix, bone and structures of interest the paintbrush tool was used for manual segmentation. 3D surface models and volumes were created to visualize the segmented structures and to illustrate this article with traditional figures. In addition, surface models of the individual structures were downsampled to a degree that allowed for small file sizes but preserved all details, and were exported as separate OBJ files for the creation of the interactive 3D figures in the supplementary material as outlined in *Lautenschlager* (*2014*) using Adobe 3D reviewer (Adobe Systems Inc.).

As both taxa have been compressed dorsoventrally to a moderate amount, the resulting endocasts were retrodeformed. For the retrodeformation process, the digital skull and endocast models were scaled in dorsoventral direction using the "transform editor" in Avizo. BSPG 1931 X 502 was scaled to the same dorsoventral dimensions as a less compressed skull of *Parasuchus hislopi* (ISI R42, Indian Statistical Institute, Kolkata, India), corresponding to a scaling factor of approximately 40%. The same scaling factor was assumed for BSPG 1931 X 501 based on the fact that both specimens were found in close proximity to one another on a single bedding plane and likely had a similar diagenetic history (*Butler et al.*, *2014*).

## RESULTS

### Endocranial anatomy

The endocasts of *Ebrachosuchus neukami* and *Parasuchus angustifrons* are very similar in their morphology. Both endocasts are elongate, straight (i.e. arranged horizontally) and mediolaterally narrow (Figs. 2 and 3). Long olfactory tracts extend rostrally and are as long as the main portion of the endocasts in each taxon. Fossae for olfactory bulbs are preserved in both taxa, but only in *Parasuchus angustifrons* is a rostral separation into two olfactory bulbs visible (Fig. 2B). The ventral extent could not be reconstructed as no bony structures cover this region. The cerebrum and the cerebral hemispheres are only weakly demarcated and form the widest part of the endocast in each taxon. Again, in *Parasuchus angustifrons* the cerebral hemispheres are slightly more prominent than in *Ebrachosuchus neukami*. The mid- and hindbrain region, including the cerebellum, is mediolaterally compressed between the endosseous labyrinths in both taxa. Caudally, the endocasts increase in width towards the foramen magnum. The floccular lobes, extending from the cerebellum, are prominent but short. In comparison to the more oval-shaped morphology in *Parasuchus angustifrons*, the floccular lobes are slightly dorsoventrally flattened in *Ebrachosuchus*
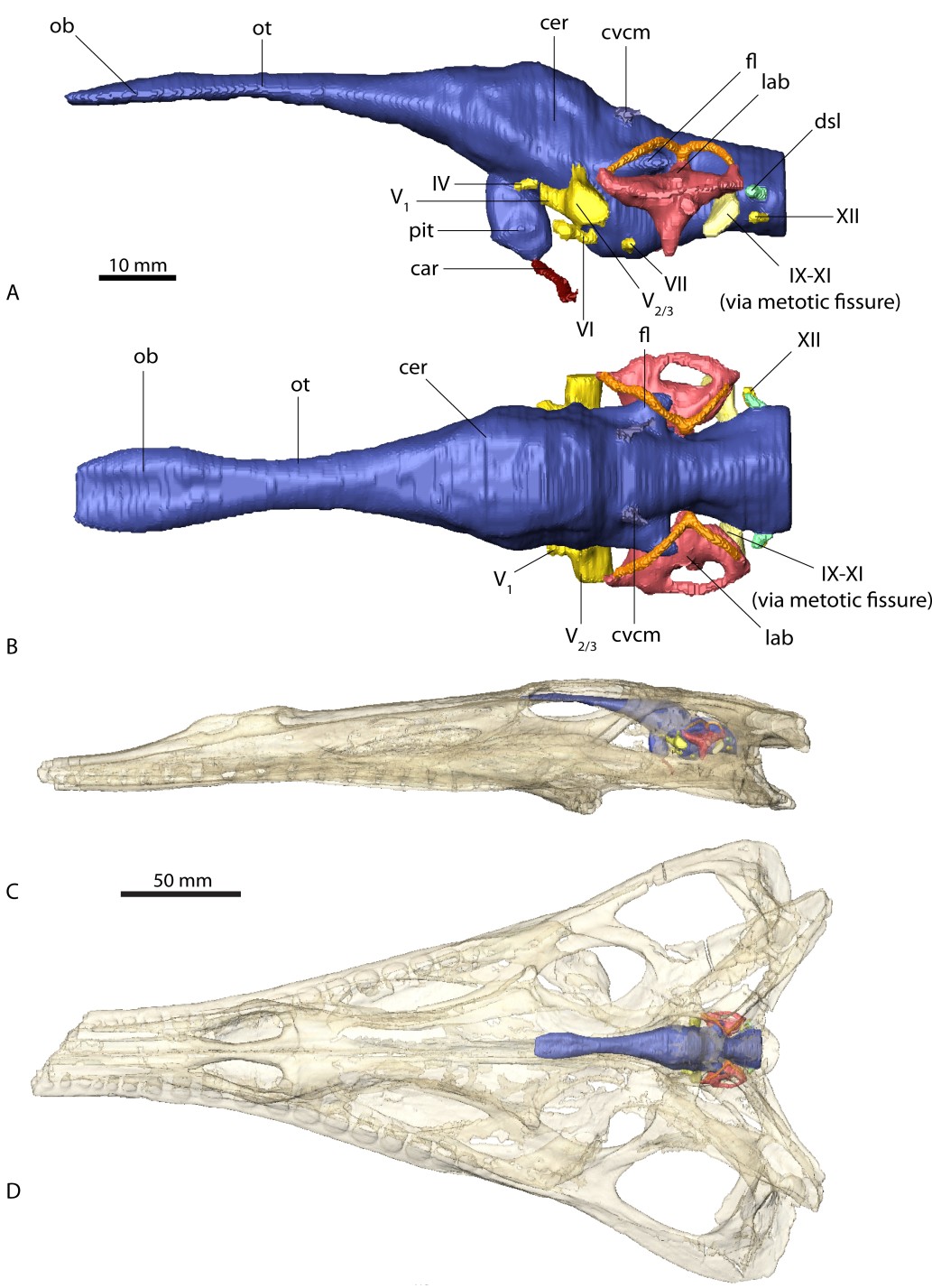

**Figure 2  Endocranial anatomy of *Parasuchus angustifrons* (BSPG 1931 X 502).** Endocast of brain and endosseous labyrinth in (A) left lateral and (B) dorsal view. Endocast *in situ* in (C) left lateral and (D) dorsal view with bone rendered semi-transparent.

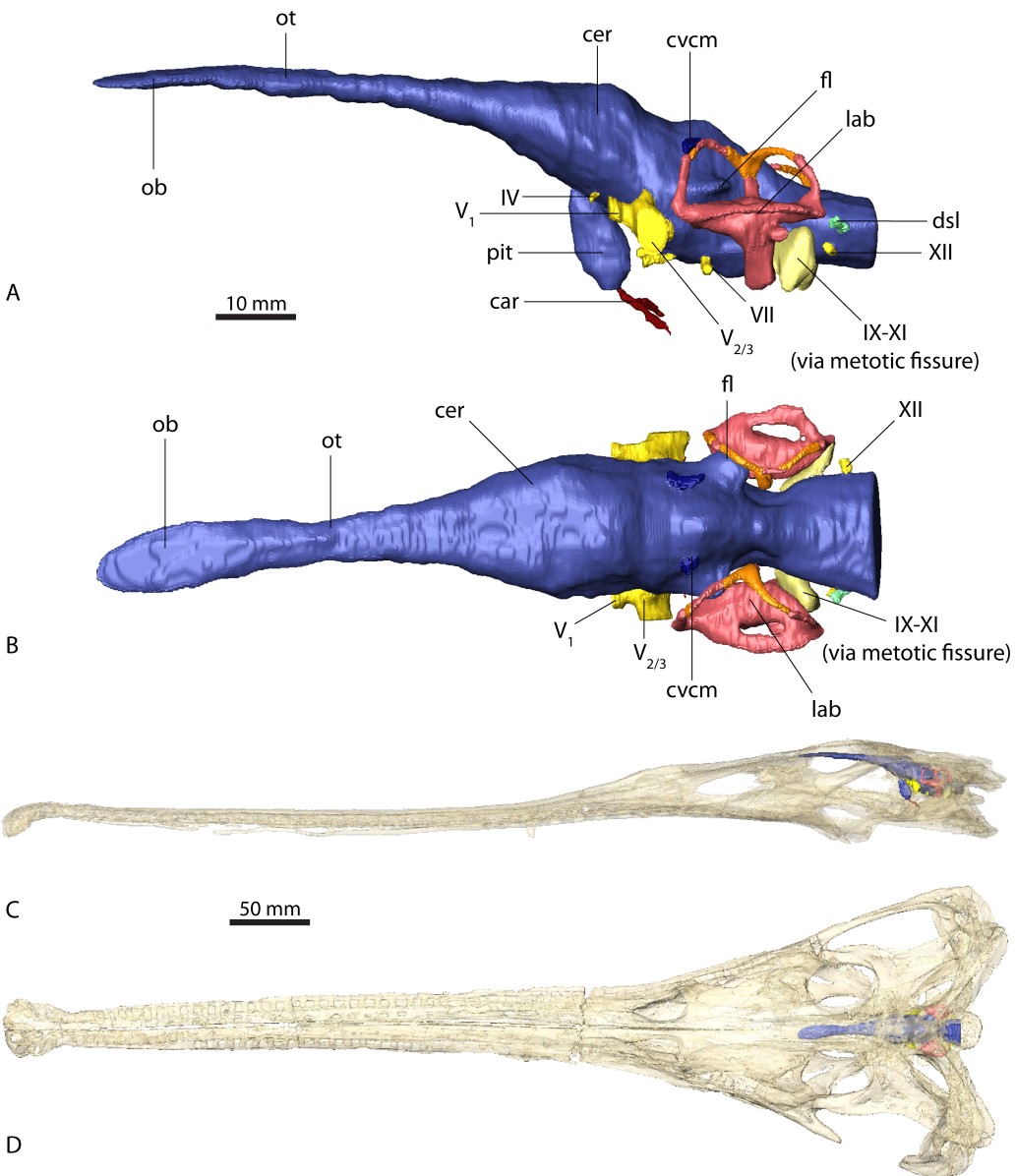

**Figure 3 Endocranial anatomy of *Ebrachosuchus neukami* (BSPG 1931 X 501).** Endocast of brain and endosseous labyrinth in (A) left lateral and (B) dorsal view. Endocast *in situ* in (C) left lateral and (D) dorsal view with bone rendered semi-transparent.

*neukami* (Fig. 3A). In both taxa, the floccular lobes enter the vestibular apparatus of the endosseous labyrinth, but do not extend beyond the rostral semicircular canal. Despite the elongate morphology of the endocasts, the cephalic flexure (between the fore- and mid-brain) and the pontine flexure (between the mid- and hindbrain) are pronounced in the endocasts. The cast of the pituitary fossa is prominent and extends ventrally from the ventral surface of the cerebrum. The pituitary fossae have equal dimensions and positions in both taxa.

The canals of the neurovascular structures were reconstructed for as far as indicated by osteological correlates. Their arrangement and dimensions are similar in *Ebrachosuchus neukami* and *Parasuchus angustifrons*. The optic nerve (CN II) and the oculomotor nerve (CN III) canals could not be traced due to the lack of preserved bony structures on the rostroventral part of the endocasts. The trochlear nerve canal (CN IV) is small and originates from the ventral surface of the cerebrum, rostral to the trigeminal nerve (CN V). The latter is prominent in both taxa and originates from the ventrolateral surface of the midbrain region. A split into a rostrally directed ophthalmic branch (CN $V_1$) and a laterally projecting combined canal for the maxillary (CN $V_2$) and mandibular (CN $V_3$) branches is evident in both endocasts. A subdivision of the latter two is not visible in the CT scans, but most likely occurred further outside of the endocranial cavity as is the plesiomorphic archosaurian condition (*Witmer et al.*, *2008*). The canal for the abducens nerve (CN VI) originated from the ventral surface of the endocast below the trigeminal nerve canal. The canal for the facial nerve (CN VII) is situated caudal to that of the abducens nerve. The vestibulocochlear nerve canal (CN VIII) could not be reconstructed in either taxon as the resolution of the CT scans is not clear enough in this region to identify the nerve canal confidently. A large metotic fissure is present in both taxa, transmitting the glossopharyngeal (CN IX), the vagus (CN X) and the spinal accessory nerves (CN XI). The hypoglossal nerve (CN XII) exits the braincase via a single nerve canal. A further foramen located dorsal to the hypoglossal nerve foramen has a blind ending and likely represents a diverticulum of the longitudinal sinus (*Witmer & Ridgely*, *2008*; *Witmer & Ridgely*, *2009*).

Due to the resolution of the CT data set only the larger vascular structures could be reconstructed. The roots of the caudal middle cerebral vein are prominent and originate from the cerebellum rostrodorsally to the floccular lobes in *Ebrachosuchus neukami* and *Parasuchus angustifrons*. They can be traced caudally through the bone for a short extent exiting the braincase near the supraoccipital-parietal suture. Ventrally, the canals for the carotid artery originate from the pituitary fossa and exit the basisphenoid ventrolaterally.

In comparison to the other endocranial components, the endosseous labyrinths of *Ebrachosuchus neukami* and *Parasuchus angustifrons* show more prominent differences (Fig. 6). In general, the labyrinths are dorsoventrally short and compact. The vestibular apparatus approaches a rectangular outline and is rostrocaudally elongate in both taxa, but more pronounced in *Ebrachosuchus neukami*. This may partly due to the preservation of *Parasuchus angustifrons*, which seems to have been dorsoventrally compacted to a moderate extent (Figs. 6A and 6B). In *Ebrachosuchus neukami*, the rostral semicircular canal is the longest and describes a somewhat quadrangular shape, whereas the caudal semicircular canal is more oval-shaped (Figs. 6C and 6D). The lateral semicircular canal is short and compact. The semicircular canals in *Parasuchus angustifrons* appear, as far as preserved, dorsoventrally compressed and with more uniform dimension than in *Ebrachosuchus neukami*, although this is partly a preservational artefact. The cochlear ducts are short in *Ebrachosuchus neukami* and *Parasuchus angustifrons* and extend largely ventrally, with only a slight medial component. The fenestra vestibuli were reconstructed in *Ebrachosuchus neukami* and *Parasuchus angustifrons*.

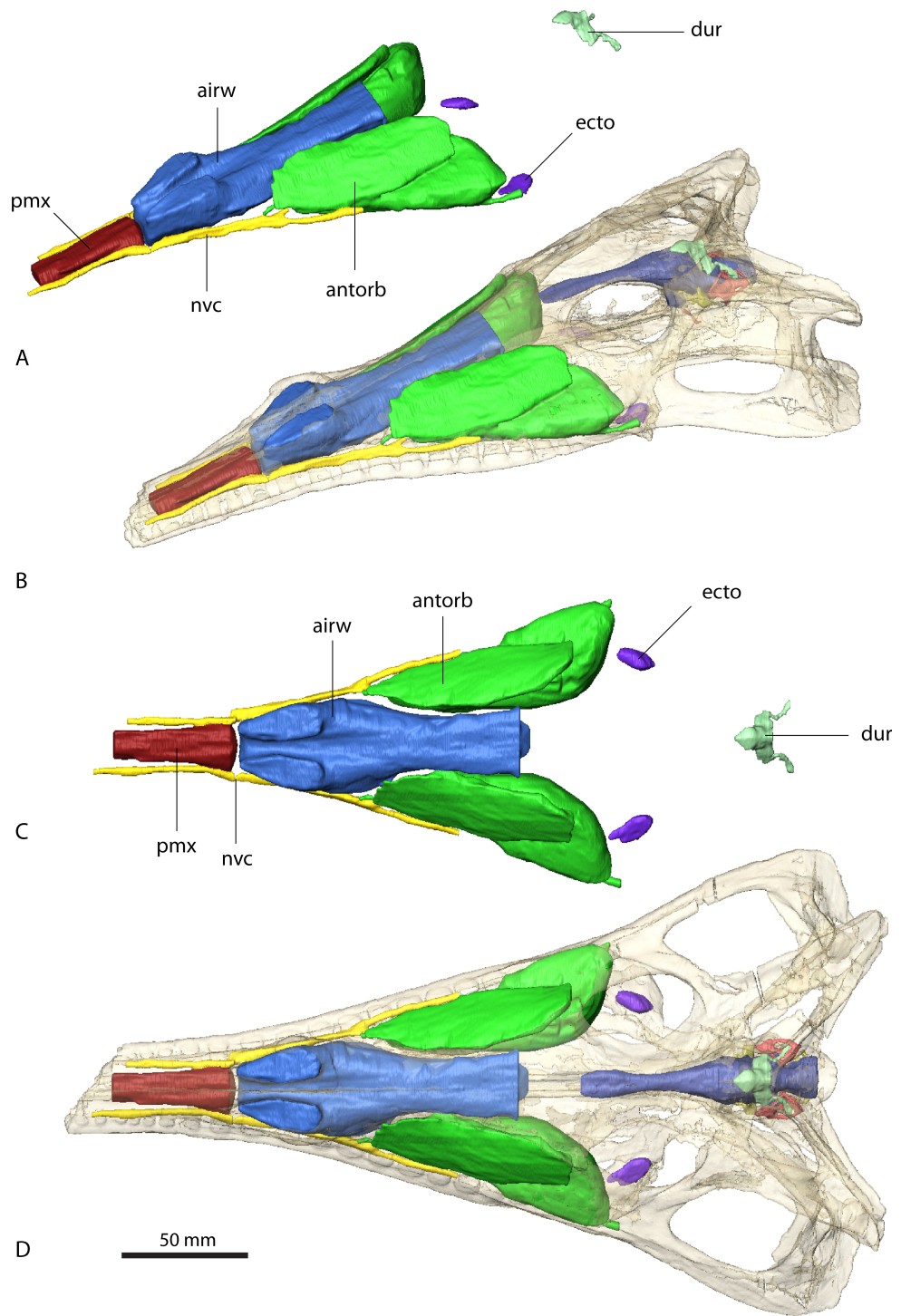

**Figure 4** **Paranasal sinuses of *Parasuchus angustifrons* (BSPG 1931 X 502).** Sinuses in (A) and (B) in rostrolateral view and (C) and (D) dorsal view. Sinuses in (B) and (D) *in situ* with bone rendered semi-transparent.

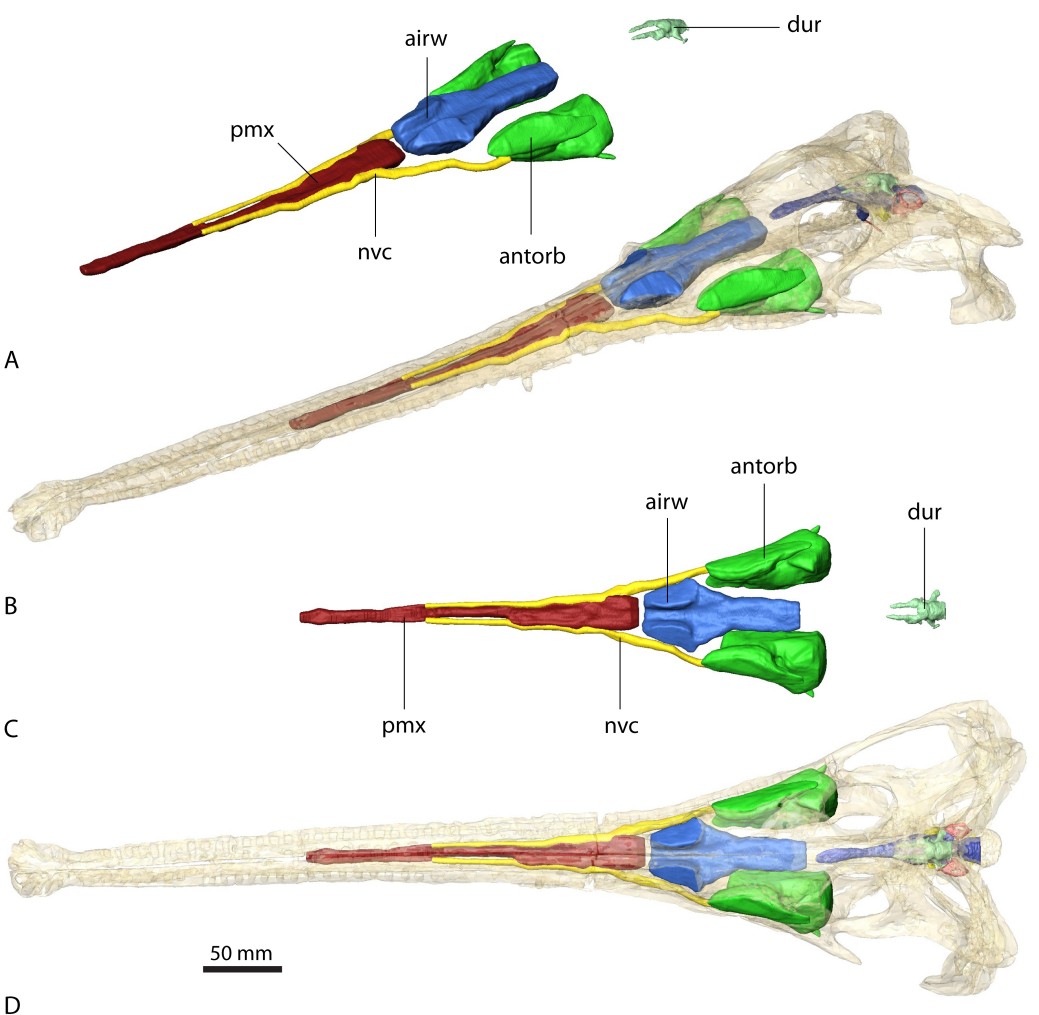

**Figure 5** **Paranasal sinuses of *Ebrachosuchus neukami* (BSPG 1931 X 501).** Sinuses in (A) and (B) in rostrolateral view and (C) and (D) dorsal view. Sinuses in (B) and (D) *in situ* with bone rendered semi-transparent.

As with the endocranial cavities, the arrangement and extent of the various sinuses is similar in *Ebrachosuchus neukami* and *Parasuchus angustifrons* (Figs. 4 and 5). The antorbital sinus is large and fills the antorbital fenestra, as well as the space between the palate and the palatal shelf of the maxilla. A small diverticulum also appears to enter the jugal via a foramen near the ectopterygoid-jugal contact (*Butler et al.*, *2014*) in both taxa. Rostrally, a large canal is present, which opens into the antorbital cavity. The canal likely transmitted neurovascular structures, including the maxillary branch of the trigeminal nerve. The region rostral to the external nares comprises a large air-filled space for the entire length of the premaxilla medial to the neurovascular canal supplying the alveolar cavities. This region may have housed a premaxillary sinus (most likely as extension of the antorbital sinus) and/or neurovascular bundles (*Butler et al.*, *2014*) as in extant crocodiles (*Leitch & Catania*, *2012*). The airway is simple and unbranched in *Ebrachosuchus neukami*

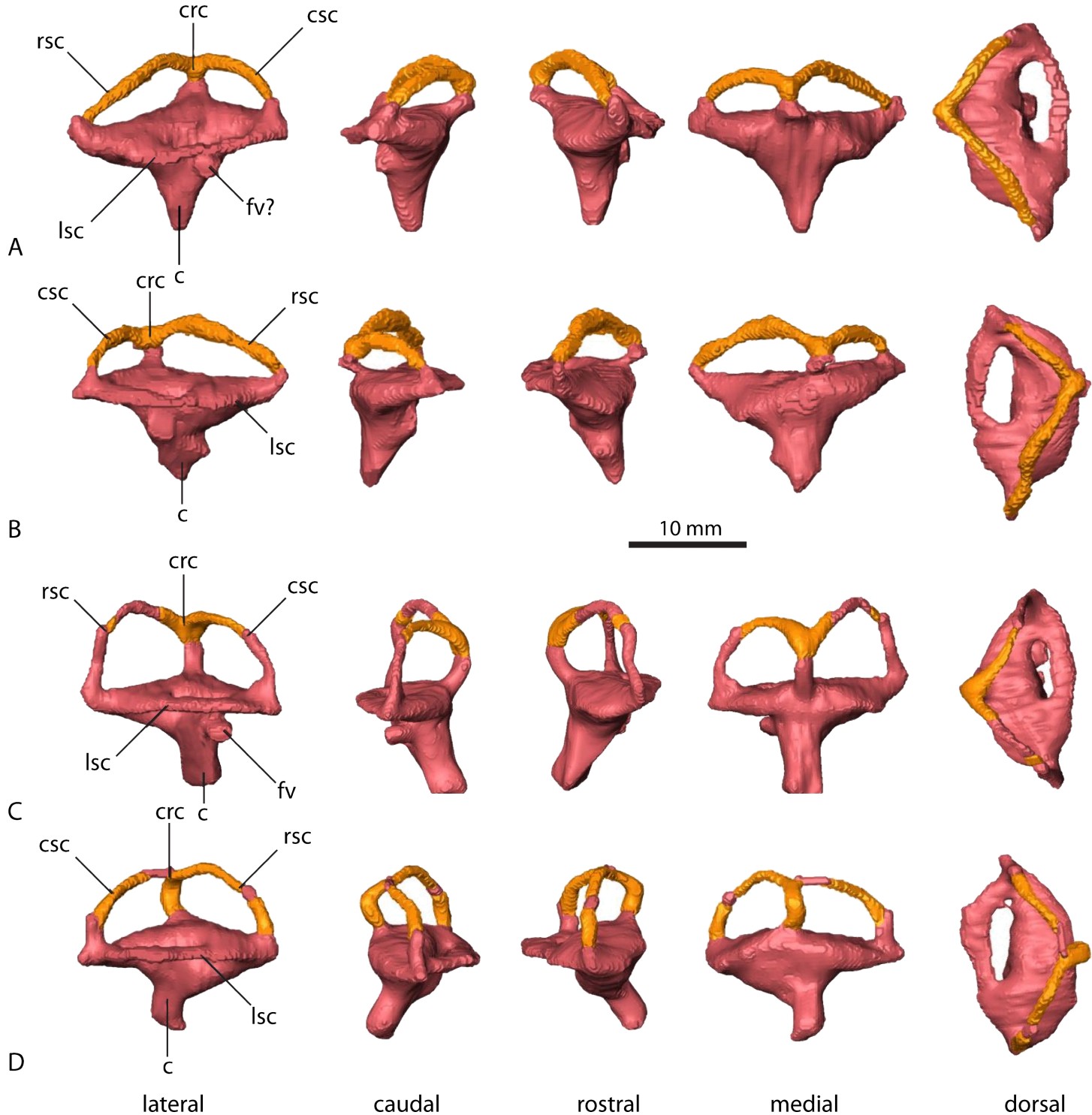

lateral        caudal        rostral        medial        dorsal

**Figure 6** **Endosseous labyrinths.** *Parasuchus angustifrons* (BSPG 1931 X 502) (A) left labyrinth, (B) right labyrinth. *Ebrachosuchus neukami* (BSPG 1931 X 501) (C) left labyrinth, (D) right labyrinth. Parts reconstructed and reflected from the opposite side (where preserved) shown in different colour.

and *Parasuchus angustifrons* and connects the external nares with the choanae and the olfactory region.

In *Parasuchus angustifrons*, the ectopterygoid is pierced medially by a single oval foramen opening into a large cavity within the bone (Fig. 4). No further foramina are identifiable, suggesting the foramen connected to a pneumatic recess rather than transmitting neurovascular structures. The pneumatic recess is either part of the antorbital sinus or a separate ectopterygoid sinus of unknown source (*Witmer*, *1997*). The respective region is only partly preserved and damaged, but a similarly large cavity appears to be absent in *Ebrachosuchus neukami*. A further sinus located dorsal to the brain endocast is present in *Ebrachosuchus neukami* and *Parasuchus angustifrons*. While this structure could be interpreted as part of the paratympanic sinus, there is no clear connection to the middle ear visible in the datasets. In both taxa, it covers the cerebellum dorsally. In *Ebrachosuchus neukami* two small diverticula extend rostrally covering the cerebrum dorsolaterally. These diverticula are not visible in *Parasuchus angustifrons*. Laterally, subsidiary canals of the tympanic sinus are present in both taxa, but more pronounced in *Parasuchus angustifrons*, in which they exit the braincase via a foramen between the parietal and the prootic and connect to the caudal tympanic recess. This sinus possibly had a further connection to the quadrate foramen, but the pathway for this canal is not indicated by osteological correlates. The sinus is therefore most likely a combination of the endocranium and the dural venous sinuses.

## Comparison with other phytosaurs

A comparison with other phytosaurs shows that, while similar to each other, the endocranial anatomy of *Ebrachosuchus neukami* and *Parasuchus angustifrons* differs in several aspects from that of more derived taxa (Fig. 7). However, it should be noted that accurate comparisons are exacerbated by the scarcity of detailed reconstructions of endocasts. Existing reconstructions are mostly based on physical casts (natural and artificial) or interpretive drawings (*Cope*, *1888*; *Case*, *1928*; *Mehl*, *1928*; *Camp*, *1930*; *Chatterjee*, *1978*). All phytosaur endocasts appear to share a basic bauplan with the individual brain regions arranged longitudinally (in contrast to a more vertical arrangement such as seen in birds) and a mediolaterally narrow morphology. The olfactory tracts are significantly elongate in all taxa (as far as preserved/reconstructed), making up approximately half the length of the entire endocasts. Caudal to the olfactory tracts, the various taxa show large differences in the orientation of the individual brain portions. Cephalic and pontine flexure is very variable. While *Ebrachosuchus neukami* and *Parasuchus angustifrons* share very large flexure angles (following *Lautenschlager & Hübner*, *2013*) with derived taxa, such as *Machaeroprosopus pristinus* and *Machaeroprosopus buceros*, the fore- and mid-brain and the mid- and hind-brain appear to be almost perpendicular to each other in *Smilosuchus gregori* and *Parasuchus hislopi*. However, although the studied specimens of *Parasuchus angustifrons* (BSPG 1931 X 502) and *Ebrachosuchus neukami* (BSPG 1931 X 501) are well preserved and mostly complete, they show signs of moderate dorsoventral compaction. Retrodeformed endocast reconstructions exhibit cephalic and pontine flexures more similar to *Machaeroprosopus mccauleyi* and *Parasuchus hislopi* (Fig. 8). A significant difference is

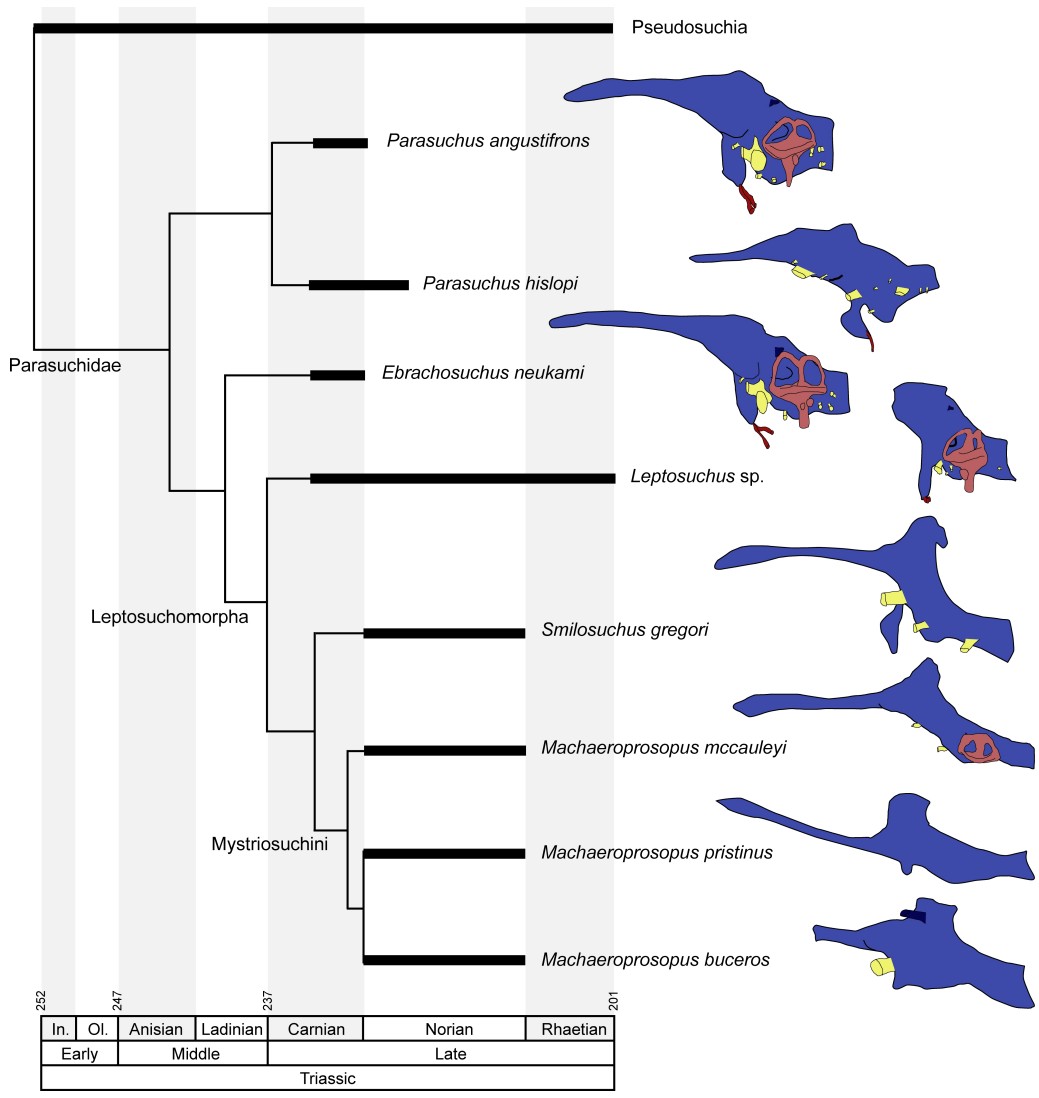

**Figure 7** **Endocranial anatomy of different phytosaurian taxa.** Comparisons based on existing endocasts and endocast reconstruction redrawn from *Cope* (*1888*), *Case* (*1928*), *Mehl* (*1928*), *Camp* (*1930*), *Chatterjee* (*1978*) and *Holloway, Claeson & O'Keefe* (*2013*). Time-calibrated phylogeny based on *Stocker & Butler* (*2013*), *Kammerer et al.* (*2016*) and *Ezcurra* (*2016*). Endocasts of *Parasuchus angustifrons* and *Ebrachosuchus neukami* shown after retrodeformation.

found in the presence of a pineal organ or epiphysis dorsal to the cerebrum. A pineal organ has been suggested to be present (*Jaekel*, *1910*; *Langston*, *1949*) and been reconstructed for the majority of phytosaurs, but is absent in *Ebrachosuchus neukami* and *Parasuchus angustifrons*. The dorsal expansion in the respective region in these taxa is interpreted in this study to represent parts of the dural venous sinus or alternatively the paratympanic sinus, due to the rostral and lateral expansion of this structure into parts of the braincase. *Hopson* (*1979*) similarly considered a pineal organ in phytosaurs unlikely and suggested that the respective region in the endocranial cavity housed a cartilaginous portion of the supraoccipital. Although reconstructed by *Mehl* (*1928*), an enlarged epiphysis was reported

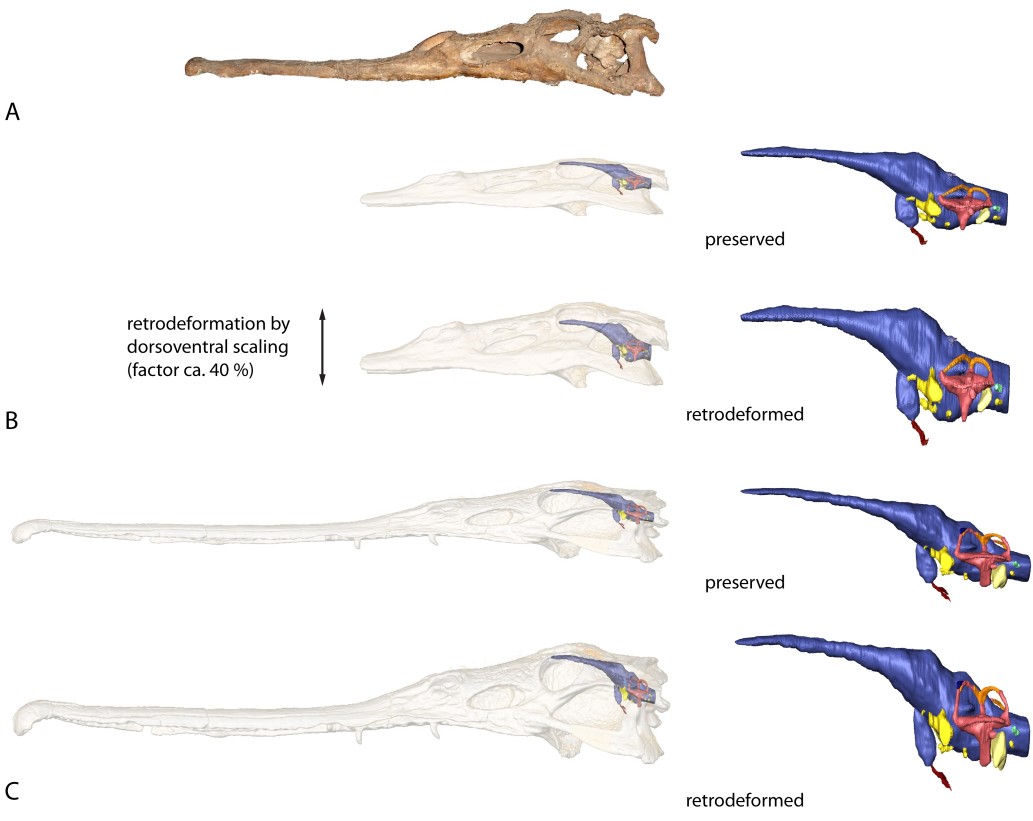

**Figure 8** **Retrodeformation of studied taxa.** (A) Complete and undistorted skull of *Parasuchus hislopi* (ISI R42) used as a guide for retrodeformation of (B) *Parasuchus angustifrons* (BSPG 1931 X 502) and (C) *Ebrachosuchus neukami* (BSPG 1931 X 501).

to be absent in *Machaeroprosopus pristinus* in an as-yet-unpublished recent study (*Smith et al., 2010*). Existing endocast reconstructions provide ambiguous results regarding the presence of the epiphysis, but suggest that it may have elaborated through phytosaur evolution (Fig. 7). However, considering its absence in modern crocodilians, the epiphysis must have been lost at some stage prior to the origin of the crocodilian crown group.

Similar to the actual brain endocast, the endosseous labyrinths show, as far as reconstructed, subtle differences between different phytosaurian taxa. The vestibular part of the labyrinth of *Ebrachosuchus neukami* and *Parasuchus angustifrons* is rostrocaudally longer than dorsoventrally high, whereas it seems to have more uniform dimensions in *Machaeroprosopus mccauleyi* and *Leptosuchus* sp. Again, this is to some extent a result of dorsoventral compression. The retrodeformed endosseous labyrinths of *Ebrachosuchus neukami* and *Parasuchus angustifrons* are more similar to the labyrinth of *Leptosuchus* sp. It should be noted, though, that the retrodeformation is based on dorsoventral scaling of the complete skull and endocast, respectively. However, the opisthotic and the paroccipital, which house the endosseous labyrinth, might not have suffered the same amount of compression as the complete skull. Furthermore, the scarcity of reconstructed and preserved natural labyrinthine endocasts confounds wider comparisons.

## Comparison with Crocodyliformes

As in comparison with other phytosaurs, the endocranial anatomy of *Ebrachosuchus neukami* and *Parasuchus angustifrons* shares a basic bauplan with most Crocodyliformes in the form of a longitudinally arranged and elongate brain architecture thought to be plesiomorphic for the whole lineage (*Hopson*, *1979*). As in phytosaurs, the olfactory tracts are elongate in extant crocodylians, including *Alligator mississippiensis* (*Witmer & Ridgely*, *2008*), *Crocodylus johnstoni* (*Witmer et al.*, *2008*) and *Crocodylus moreleti* (*Franzosa*, *2004*), as well as in several phylogenetically distinct Mesozoic longirostrine crocodylomorphs, such as the neosuchian *Pholidosaurus* (*Edinger*, *1938*; *Hopson*, *1979*) and the metriorhynchid *Cricosaurus araucanensis* (*Herrera, Fernandez & Gasparini*, *2013*). In contrast to phytosaurs, the cerebral hemispheres are prominent and mediolaterally enlarged in most Crocodyliformes (*Wharton*, *2000*; *Franzosa*, *2004*; *George & Holliday*, *2013*). Extant crocodylians possess an enlarged dural venous sinus covering the endocast dorsally (*Witmer et al.*, *2008*), which has been interpreted to be present in fossil Mesoeucrocodylia (*Hopson*, *1979*; *Wharton*, *2000*). Where preserved or reconstructed, the endosseous labyrinths show a dorsoventrally compressed vestibular region and short cochlear ducts in Crocodyliformes (*Franzosa*, *2004*; *Witmer et al.*, *2008*), similar to *Ebrachosuchus neukami* and *Parasuchus angustifrons*.

Paranasal sinuses have been reconstructed only for a handful of extant and extinct Crocodyliformes (e.g., *Alligator mississippiensis*, *Cricosaurus araucanensis*), which limits comparisons of these structures. A clear difference is found in the size of the antorbital sinus. In *Ebrachosuchus neukami* and *Parasuchus angustifrons* the antorbital sinus is enlarged, but it is considerably smaller in Crocodyliformes (*Witmer & Ridgely*, *2008*; *Herrera, Fernandez & Gasparini*, *2013*). Due to the position of the external nares, the airway is short in the studied phytosaurs. In the longirostrine metriorhynchid *Cricosaurus araucanensis*, the rostrum comprises the airway for its entire length (*Herrera, Fernandez & Gasparini*, *2013*), whereas the comparable region was likely filled by a premaxillary sinus in phytosaurs.

## DISCUSSION

The reconstruction of the endocranial anatomy of *Ebrachosuchus neukami* and *Parasuchus angustifrons* suggests that the general bauplan of pseudosuchian brain architecture was already established in Phytosauria. Plesiomorphic characters, such as elongate olfactory tracts, a mediolaterally narrow and serially aligned brain and a comparably small cerebral region, are largely retained in other phytosaurs, but also in most Crocodyliformes. In contrast, features that occur in the evolution of avemetatarsalian archosaurs such as a rearrangement of the brain architecture, a hyperinflated cerebrum and a reduction of the olfactory apparatus (*Zelenitsky et al.*, *2011*; *Balanoff et al.*, *2013*) are absent in the pseudosuchian lineage.

However, in spite of these overall similarities there are a number of differences present in the endocranial anatomy when comparisons are made between various phytosaurian taxa, but also in comparison to the (admittedly small number of) available endocranial reconstructions of Crocodyliformes. Whether these reflect subtle ecological or behavioural

adaptations, intraspecific variation or interpretive artefacts is difficult to discern. The small sample size and lack of detailed, three-dimensional reconstructions currently prevents rigorous tests of the latter two possibilities. It is generally assumed that the osteological similarities between phytosaurs and longirostrine Crocodyliformes are the result of convergent evolution and the adaptation to the same habitat and/or diet (e.g., *Camp*, *1930*; *Hunt*, *1989*). Similarities or differences in the endocranial anatomy could therefore indicate adaptive changes of key structures. Apart from the plesiomorphic morphology of the brain inherent to both phytosaurs and Crocodyliformes, both groups share a dorsoventrally flattened and rostrocaudally expanded morphology of the vestibular apparatus of the inner ear. Such an increase in the aspect ratio of the vertical semicircular canals has been associated with an adaptation to an aquatic environment (*Georgi & Sipla*, *2008*) and is found also in other marine reptiles (*Neenan & Scheyer*, *2012*). It is therefore possible that the endocranial anatomy in phytosaurs and longirostrine Crocodyliformes follows a shared plesiomorphic pattern that has been convergently modified in response to similar sensory adaptations. Additional sampling of phytosaur and fossil crocodyliform endocasts and more refined palaeobehavioural and palaeoecological data will be required to provide a more definitive assessment of this hypothesis.

## CONCLUSIONS

The digital reconstruction of the brain, inner ear, neurovascular and sinus morphology of the two non-mystriosuchine phytosaurs *Parasuchus angustifrons* and *Ebrachosuchus neukami* offers new insights into the endocranial anatomy and evolution of Phytosauria. The endocasts of both taxa are very similar to each other in their rostrocaudally elongate morphology, with long olfactory tracts, weakly demarcated cerebral regions and dorsoventrally short endosseous labyrinths. Several sinuses, including large antorbital sinuses and prominent dural venous sinuses, were reconstructed. Comparisons with published endocranial reconstructions of other, more derived, phytosaurian taxa demonstrate a substantial morphological variability, most pronounced in the cephalic and pontine flexure and the presence of a pineal organ. Endocranial characters that are found across all phytosaurs, as far as preserved, include the elongate olfactory tract and a serially arranged brain architecture. As far as allowed by the limited available comparative data, these features appear to be shared with members of the clade Crocodyliformes. However, the scarcity of reconstructed endocasts for phytosaurs and crocodyliforms, as well as preservational artefacts, confound large-scale comparisons and provide an impetus for further future work on the endocranial anatomy and evolution of these clades.

### Anatomical abbreviations

| | |
|---|---|
| **airw** | airway |
| **antorb** | antorbital sinus |
| **c** | cochlear duct |
| **car** | carotid artery |
| **cer** | cerebral hemisphere |
| **crc** | crus communis |

| csc | caudal semicircular canal |
| cvcm | caudal middle cerebral vein |
| dsl | diverticulum of longitudinal sinus |
| dur | dural venous sinus |
| ecto | ectotympanic sinus |
| fl | floccular lobe |
| fv | fenestra vestibuli |
| lab | endosseous labyrinth |
| lsc | lateral semicircular canal |
| nvc | neurovascular canal |
| ob | olfactory bulbs |
| ot | olfactory tracts |
| pit | pituitary fossa |
| pmx | premaxillary sinus |
| rsc | rostral semicircular canal |
| IV | trochlear nerve canal |
| $V_1$ | ophthalmic branch of the trigeminal nerve canal |
| $V_2$ | maxillary branch of the trigeminal nerve canal |
| $V_3$ | mandibular branch of the trigeminal nerve canal |
| VI | abducens nerve canal |
| VII | facial nerve canal |
| IX–XI | shared canal for the glossopharyngeal, vagus and spinal accessory nerve |
| XII | hypoglossal nerve canal |

## ACKNOWLEDGEMENTS

We thank Martin Dobritz (Klinikum rechts der Isar, Munich) for CT scanning and Oliver Rauhut (BSPG, Munich) for access to specimens. We thank Jon Tennant (Imperial College London) and Ryan Ridgely (Ohio University) for their helpful reviews of an earlier version of this manuscript.

### Funding

RJB's research was supported by an Emmy Noether Programme Award from the Deutsche Forschungsgemeinschaft (BU 2587/3-1) and a Marie Curie Career Integration Grant (PCIG14-GA-2013-630123 ARCHOSAUR RISE). The funders had no role in study design, data collection and analysis, decision to publish, or preparation of the manuscript.

### Grant Disclosures

The following grant information was disclosed by the authors:
Deutsche Forschungsgemeinschaft: BU 2587/3-1.
Marie Curie Career Integration: PCIG14-GA-2013-630123 ARCHOSAUR RISE.

## Competing Interests

The authors declare there are no competing interests.

## Author Contributions

- Stephan Lautenschlager conceived and designed the experiments, performed the experiments, analyzed the data, wrote the paper, prepared figures and/or tables, reviewed drafts of the paper.
- Richard J. Butler conceived and designed the experiments, analyzed the data, contributed reagents/materials/analysis tools, wrote the paper, reviewed drafts of the paper.

## Data Availability

CT data sets are deposited with the specimens in the BSPG (Bayerische Staatssammlung für Paläontologie und Geologie, Munich, Germany) collections. In addition, DICOM files have been deposited on figshare:

Parasuchus: 10.6084/m9.figshare.3443963, https://figshare.com/s/06c424a37261bc973be8

Ebrachosuchus: 10.6084/m9.figshare.3443960, https://figshare.com/s/aa3e27406b9fd02352dc.

## Supplemental Information

Supplemental information for this article can be found online at http://dx.doi.org/10.7717/peerj.2251#supplemental-information.

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
