# Peer review of "Neural and endocranial anatomy of Triassic phytosaurian reptiles and convergence with fossil and modern crocodylians"

_PeerJ, doi:10.7717/peerj.2251_

## Round 0.1 · original submission · Minor Revisions

Both reviewers have stressed the good quality of this study and encouraged its publication. However, they have also made very important remarks that need to be taken into full consideration before we consider its publication.

Please also note the 2 attachments from the reviewers.

·

Basic reporting

Figures
The figures are well annotated, numerous, and informative. I strongly recommend that these be used in a press release for this paper, seeing as it pushes our boundary about what we know about phytosaurs in an exciting way.

Data
See comments below in main response file.

The reporting for this manuscript is to a very high scientific standard in all other aspects.

Experimental design

The study presents new research, and is well within the scope of PeerJ. The research question is well-defined, but could do with a bit more information on the scope and importance of the study. The technical standard of work performed is very high. If the scans are deposited in a more public repository, then this study should be broadly replicable (this is somewhat unclear at the present, see comments below). The title might also be slightly misleading, as it refers to Crocodylia while the study is broader by investigating Crocodyliformes.

Validity of the findings

The data collected in this study is of high quality. No statistical tests were performed. The conclusions are broadly supported by the data, but a couple of caveats/limitations could be added for increased context.

Additional comments

Please note that a full review is provided in the attachment file.

·

Basic reporting

The manuscript's introduction is extremely well done, which made this paper much easier to review. Figures and CT data processing are exceptional, as expected from these authors. The raw CT data do need to be made available online, if possible.

Experimental design

This work is descriptive anatomy from historical biology. As such, it meets all the standards for reporting in this discipline to a very high degree.

Validity of the findings

Most of the findings are solid, however, I question some anatomical interpretations from the CT data.

Additional comments

Excellent contribution that fills a large gap in our understanding in phytosaur evolution.

Lines 88-89: Might be worth mentioning that the deformation is is almost entirely plastic, with no apparent brittle deformation.

Lines 106-108: Which tool in Avizo was used? Technically, the models were scaled up dorsoventrally. “Non-uniformly” adds confusion.

Line 127: Confusing wording. Perhaps: “Fossae for olfactory bulbs were preserved”?

Line 158: A large "metotic fissure" can be seen in the CT data for both taxa housing CN IX-XI. This can also be seen on Parasuchus hislopi (ISI R44) in Chatterjee 1978.

Line 163-164: Going through the data, I was able to find only a single exit for CN XII. The more rostral of the two foramina the authors posit for XII is “blind” in both taxa, and thus likely corresponds to the “diverticulum of the longitudinal sinus” of Hopson, 1979. See also Witmer & Ridgely 2008, 2009.

Line 184: One of these fenestrae is an artifact of the data or the specimen.

Line 190-193: This is confusing, as it reads as if the neurovascular canal transmits a pneumatic cavity.

Line 197-198: Can the olfactory region and choana be visualized in the 3D reconstructions?

Line 200: Is there a single foramina in the ectopterygoid? In other words, could it be transmitting a neurovascular structure?

Lines 203-211: This structure could be interpreted as paratympanic, however there is no connection to the middle ear. This structure *is* continuous with the endocranium, and its “branches” exit laterally into the dorsal temporal fossae, and caudally onto the occiput. As such, this structure is more compatible with being interpreted as a combination of endocranium and dural venous sinuses (Witmer et al, 2008)

Lines 220-221: Versus what? Most vertebrates have a serially arranged brain.

Lines: 236-237: The retrodeformation exercise for Parasuchus results in a striking resemblance to P. hislopi in Chatterjee 1978. As such, I’d suggest using this reconstruction in Figure 7 as well as rotating P. hislopi to match. Chatterjee had no orientation provided in 1978, but the olfactory tracts should be roughly horizontal.

Line 243-246: A great strength of this paper is that it shows that a pineal or epiphysis *may* have been “elaborated” through phytosaur evolution towards the crown. It’s a difficult question, as the extant phylogenetic bracket fails us. That said, we don’t know when the pineal was lost in archosaur evolution towards crocs.

Line 287-290: Could dorsoventral crushing affect the interpretation of the basal morphology? In other words, would the retrodeformed models change the story?

Line 294-296: Last sentence of this paragraph is extraneous and not needed.

Figs 2 & 3: Please consider revisiting the segmentation. I’ve gone through the data briefly, and provided an image of alternative interpretations of several structures at the caudal end of both endocasts.

Figure 4 & 5: Can connections (an ostium?) be seen/made between the airway, premax sinus, and antorbital cavity? Can the olfactory region be visualized? In Fig. 5b, the premax sinus is also labeled as airway.

Figure 6: Can the dorsal views of the labyrinths be oriented such that they are “90 degrees”, as it is, they appear to be rotated 45 degrees, which makes it difficult to compare the orientation of the lateral canal relative to other taxa.

Figure 7: Fantastic! Only suggestion is to consider the retrodeformed models, and rotate P. hislopi clockwise such that the olfactory tract is horizontal.

---

## Round 0.2 · accepted · Accept

The manuscript has been corrected taking into account the important suggestions made by both reviewers. It is a great quality study and perfectly suitable for PeerJ.

*please check for typos in detal when receiving the proofs, I came across (and reported) two, at lines 182 and 340, "glossopharyngeal" and "It is therefore possible"